# Antimicrobial and Antivirulence Activities of Carvacrol against Pathogenic *Aeromonas hydrophila*

**DOI:** 10.3390/microorganisms10112170

**Published:** 2022-10-31

**Authors:** Junwei Wang, Ting Qin, Kai Chen, Liangkun Pan, Jun Xie, Bingwen Xi

**Affiliations:** 1Wuxi Fisheries College, Nanjing Agricultural University, Wuxi 214081, China; 2Key Laboratory Freshwater Fisheries and Germplasm Resource Utilization, Ministry Agriculture, Freshwater Fisheries Research Center, Chinese Academy of Fishery Sciences, Wuxi 214081, China; 3Key Laboratory of Integrated Rice-Fish Farming Ecology, Ministry of Agriculture and Rural Affairs, Freshwater Fisheries Research Center, Chinese Academy of Fishery Sciences, Wuxi 214081, China

**Keywords:** *Aeromonas hydrophila*, carvacrol, virulence, biofilm, antibiotics, *Ctenopharyngodon idella*

## Abstract

*Aeromonas hydrophila* is a ubiquitous Gram-negative opportunistic pathogen in the freshwater environment and the most common cause of bacterial septicemia in aquaculture. In this study, we investigated the impact of carvacrol, a natural monoterpenoid found in herbs, on the virulence of *A. hydrophila* in vitro and the antibacterial effect in combination with antibiotics. The minimum inhibitory concentration (MIC) and minimum bactericidal concentration (MBC) of carvacrol against *A. hydrophila* NJ-35 were 125 µg/mL and 250 µg/mL, respectively. Carvacrol could inhibit the virulence factors (biofilm, protease, exopolysaccharide, and hemolysin) of *A. hydrophila*, and the antibiofilm potential of carvacrol was further verified by microscopic detection. Transcriptional analyses showed that the gene expression of *flaB*, *ompA*, *aha*, *ahp*, *ela*, *act*, *aerA*, *AhyR*, and *hly* were marked as downregulated. The checkerboard assay results showed that carvacrol did not have an antagonistic effect in combination with antibiotics (florfenicol, enrofloxacin, thiamphenicol, or doxycycline hydrochloride) commonly used in aquaculture but possessed an additive-synergistic effect with neomycin sulfate. In vivo studies demonstrated that carvacrol protected grass carp (*Ctenopharyngodon idella*) from *A. hydrophila* infection. Our results indicated that carvacrol possessed significant anti-bacterial and anti-virulence effects on *A. hydrophila*.

## 1. Introduction

*Aeromonas hydrophila*, a common Gram-negative pathogenic bacterium, is ubiquitously dispersed in freshwater environments and is considered an important opportunistic pathogen of fish, amphibians, reptiles, and mammals. Outbreaks of bacterial septicemia in fish caused by *A. hydrophila* annually result in severe economic loss in aquaculture. Owing to the extended use of antibiotics required to control this bacterial disease, resistant strains of *A. hydrophila* have been detected to have a broad drug-resistance spectrum and high drug-resistance rate [1]. Bardhan and Abraham (2021) reported that most motile aeromonads (74.29%–94.44%) were multiple-antibiotic-resistant (MAR), with a MAR index in the range of 0.33–1.00 [2]. To control and prevent *A. hydrophila* infection, new, effective, and environmentally friendly antibiotic alternatives and strategies are needed. Recently, anti-virulence therapy has attracted much attention and been considered as an alternative to the killing of pathogens [3]. Anti-virulence therapy involves interfering with the virulence factors or virulence-associated processes of pathogens to reduce their virulence capacity. Previous studies have demonstrated that the use of anti-virulence medications in conjunction with antibiotics can boost the effectiveness of antibiotics and reduce the required dosage [4].

Phytochemicals are chemical compounds derived from plants, which exert good bacteriostatic activity and have been suggested as alternatives to antibiotics [5]. Carvacrol, the primary component of oregano essential oil, is extracted from the plant *Origanum vulgare* [6]. Studies have demonstrated that carvacrol is a broad-spectrum antibacterial drug with potent inhibitory activity against *Enterobacter cloacae*, *Escherichia coli*, *Pseudomonas aeruginosa*, *Vibrio cholerae*, *Staphylococcus aureus*, *Chromobacterium violaceum,* and *Clostridium difficile* [7,8,9,10,11,12,13]. Moreover, carvacrol has been detected to possess antioxidant, antiviral, antifungal, and anti-inflammatory effects [14,15,16,17,18].

Currently, carvacrol is used as a natural food preservative because of its antimicrobial properties. However, there is a lack of research on its use for treating diseases in fish. To explore the impact of carvacrol on pathogenic bacterium, *A. hydrophila* is critical to expand its utility in aquaculture. Here, we detected the antimicrobial activity of carvacrol and its impact on the growth, gene expression (*flaB*, *ompA*, *aha*, *ahp*, *ela*, *act*, *aerA*, *AhyR*, and *hly*), and activity of virulence factors (biofilm, protease, exopolysaccharide, and hemolysin) in *A. hydrophila*. In addition, we tested the synergistic effects of carvacrol combined with antibiotics against *A. hydrophila*.

## 2. Materials and Methods

### 2.1. Chemical Agents and Bacterial Strain

Carvacrol (>99% HPLC purity; CAS no. 499-75-2) and the antibiotics florfenicol, enrofloxacin, thiamphenicol, doxycycline hydrochloride, and neomycin sulfate were purchased from Aladdin (Shanghai, China). Carvacrol was dissolved in dimethyl sulfoxide (DMSO, Sigma) to obtain a stock solution of 20.48 mg/mL and then diluted with Luria–Bertani (LB) or sterile distilled water. The antibiotics tested in this study are approved and commonly used fishery drugs against bacterial diseases in aquaculture in China. 

*A. hydrophila* NJ-35, a common epidemic strain isolated from diseased carp and donated by Prof. Yongjie Liu (College of Veterinary Medicine, Nanjing Agricultural University), was cultured in LB medium at 28 °C [19]. 

### 2.2. Drug Sensitivity Tests

The susceptibility of *A. hydrophila* NJ-35 to carvacrol was determined using the broth microdilution method recommended by the Clinical and Laboratory Standards Institute (CLSI) [20]. The two-fold serial microdilution method was used in 96-well flat-bottomed polystyrene microtiter plates to determine the minimum inhibitory concentration (MIC) of carvacrol against *A. hydrophila* NJ-35. *A. hydrophila* (1 × 10^8^ CFU/mL) was inoculated into fresh LB broth containing different concentrations of carvacrol (0, 7.8125, 15.625, 31.25, 62.5, 125, 250, and 500 µg/mL). Negative and positive controls consisted of wells containing only LB and wells containing LB including bacteria, respectively. The plates were incubated at 28 °C for 24 h. The MIC was defined as the lowest concentration of carvacrol in the broth at which no bacterial growth was observed. The minimum bactericidal concentration (MBC) of carvacrol against *A. hydrophila* NJ-35 was determined using the plate colony-counting method. The suspension of each well (100 µL) with no visible growth was inoculated into LB agar, and colonies were counted after 24 h at 28 °C. The same amount of DMSO was added to the control group. The MIC of antibiotics against *A. hydrophila* NJ-35 used in this study was also determined according to the above methods. All experiments were performed in triplicate.

The effect of sub-MIC carvacrol on the growth of *A. hydrophila* NJ-35 was determined according to the method described in a previous study [21]. Briefly, *A. hydrophila* NJ-35 with an initial inoculum of 1 × 10^8^ CFU/mL was diluted into LB broth containing different sub-inhibitory concentrations of carvacrol (1/2 MIC, 1/4 MIC, 1/8 MIC, 1/16 MIC, 1/32 MIC, and 1/64 MIC) before incubating the cultures for 24 h at 28°C with continual shaking (180 rpm/min). LB medium containing 1% DMSO was used as the negative control, and LB broth without carvacrol was used as the blank control. The absorbance of the culture at 600 nm was measured every 2 h using a Multiskan GO spectrophotometer. The growth experiments were repeated thrice.

### 2.3. Synergistic Effect Assay

The synergistic action of carvacrol and selected antibiotics was tested by the checkerboard assay [22]. Florfenicol (FLF), enrofloxacin (ENF), doxycycline hyclate (DOH), thiamphenicol (THM), and neomycin sulfate (NES) are commonly permitted antibiotics used in aquaculture practices. The concentrations of carvacrol and antibiotics used in the study were set from their two MIC values and were serially diluted in two-fold steps (1/16 MIC, 1/8 MIC, 1/4 MIC, 1/2 MIC, 1 MIC). All tests were performed in triplicate. The synergistic action of carvacrol and antibiotics was detected by checkerboard assay and calculation of the FIC (fractional inhibitory concentration) index. The FIC index values were interpreted according to previous studies [23,24,25]: synergism (FICI ≤ 0.5), additivity (0.5 < FICI ≤ 1), indifferent (1 < FICI ≤ 2), and antagonism (FICI > 2). The test results were also shown with isobolograms generated with synergistic concentrations of carvacrol and antibiotics [26].

### 2.4. Biofilm Production Assay

The antibiofilm activity of carvacrol against *A. hydrophila* NJ-35 was evaluated using a crystal violet biofilm assay in 96-well cell polystyrene plates [27,28]. *A. hydrophila* NJ-35 in LB (total volume 200 µL) was inoculated at an initial inoculum of 1×10^8^ CFU/mL and cultured with carvacrol at final concentrations of 0, 1/64 MIC, 1/32 MIC, 1/16 MIC, 1/8 MIC, and 1/4 MIC at 28°C without shaking for 48 h. The negative control was 1% DMSO. After incubation, the suspensions were eliminated, and the well was rinsed three times with double-distilled water (ddH_2_O) and fixed for 15 min with 10% formaldehyde. The solutions were drained from the well, and it was allowed to air dry at room temperature. The biofilms were then dyed with 0.1% crystal violet for 15 min. The microplates were washed again, and 33% glacial acetic acid was finally added. The optical density (OD) of each well was measured at 570 nm using a spectrophotometer.

### 2.5. Microscopic Analysis of Biofilm Formation

To evaluate the antibiofilm potential of carvacrol, microscopic analyses were conducted following a previous study [29]. Briefly, the biofilm of *A. hydrophila* was formed on glass slides (1 cm × 1 cm) with different concentrations of carvacrol (0, 1/16 MIC, 1/4 MIC, and 1 MIC). After incubation, the planktonic cells were eliminated using distilled water. The glass slides were air-dried and stained with 0.5% crystal violet for 5 min. The excess stain was rinsed with distilled water. Biofilms on glass slides were examined under 400× and 1000× magnification and before photographing with a digital camera.

### 2.6. Biofilm Eradication Assay

A biofilm eradication assay was conducted following the method described in a previous study [30]. Briefly, *A. hydrophila* NJ-35 (1 × 10^8^ CFU/mL) was introduced into LB plates and incubated at 28°C for 24 h. After 24-hour incubation, phosphate-buffered saline (PBS) was used to wash the plate three times after planktonic cells had been aspirated. Then, equal volumes of LB broth containing different concentrations of carvacrol (0, 1/4 MIC, 1/2 MIC, 1 MIC, and 2 MIC) were added to the wells, while 1% DMSO was used in the negative control group. Crystal violet-staining was performed to assess the biofilm biomass at 6 h and 24 h of incubation. The OD of each well was measured at 570 nm using a spectrophotometer. All assays were performed in triplicate.

### 2.7. Exopolysaccharide (EPS) Production

The cultivation of *A. hydrophila* in 24-well plates was similar to that stated in Section 2.4. The centrifuged precipitates from the cell cultures treated with various doses of carvacrol (0, 1/16 MIC, 1/8 MIC, and 1/4 MIC) were resuspended in 10 mL buffer containing 0.85% NaCl and 0.22% formaldehyde. EPS was extracted from the aforementioned solutions by centrifugation at 15,000× *g* (4°C, 30 min) and measured using the phenol-sulfuric acid method [7].

### 2.8. Protease and Hemolysis Activity Assays

Protease activity was determined using an azocasein assay [31]. Briefly, the growth of *A. hydrophila* in LB broth with sub-inhibitory levels of carvacrol reached an OD_600_ of 0.6. The cell-free culture supernatants (CFCS) were collected by centrifugation after culture at 28 °C for 24 h. Next, 1 mL azocasein (3 mg/mL in 50 mmol/L Tris-HCl buffer, pH 8.0) was mixed with 150 µL of CFCS. After 30 min of incubation at 37 °C, 500 µL trichloroacetic acid (10%) was added to terminate the reaction. The supernatant was collected after centrifugation and neutralized with NaOH (1 mol/L). Finally, the absorbance (OD_400nm_) of the supernatant was measured.

The hemolysis activity was determined as outlined in a previous study [32]. In brief, sheep erythrocytes (4%) were centrifuged and washed with PBS (pH 7.4). Then, 100 µL CFCS were added to 900 µL fresh erythrocyte saline suspension (4%). After incubation for 30 min at 37 °C, the mixtures were centrifuged, and the absorbance (OD_540 nm_) of the supernatant (200 µL) was measured. The same volume of PBS served as the negative control, and distilled water served as the positive control (hemolysis: 100%). Hemolysis activity (%) was defined as [(OD_540nm_ sample − OD_540nm_ negative control) × 100]/OD_540nm_ positive control. All assays were performed in triplicate.

### 2.9. Quantitative Real-Time PCR 

qRT-PCR was used to assess the influence of carvacrol on gene expression of different virulence. *A. hydrophila* was treated with carvacrol (0, 1/4 MIC) for 20 h, and 1% DMSO was used as a negative control. Total RNA was extracted following the guidance and instruction of the RNAiso Plus kit (Takara, Daling, China). RNA quantities and concentrations were determined using a Nanodrop 2000 Spectrophotometer (Thermo Scientific, Waltham, MA, USA). Double-stranded cDNA was synthesized using Hiscript RT supermix for qPCR with a gDNA wiper (Vazyme, Nanjing, China). Real-time PCR was performed using SYBR green real-time PCR mix (Bio-Rad) on a CFX real-time PCR detection system (Bio-Rad, Hercules, CA, USA). The mRNA expression of targeted genes (*flaB*, *aha*, *ompA*, *ahp*, *act*, *aerA*, *hly*, *ela*, and *AhyR*) was normalized to the internal control (*rpoB* gene). Each assay was performed in triplicate. The gene-specific primers used in this study are listed in Appendix A.

### 2.10. Challenge Test

Grass carp (*Ctenopharyngodon idella*) (fish, n = 75; body weight, 50 ± 6 g) were assigned to three 500 L circular tanks with 25 fish each after a week of adapted rearing. The negative and positive control groups were fed a basal diet, while the carvacrol dietary group was fed a basal diet supplemented with 1.0 g/kg carvacrol. Fish were fed twice daily at 9 a.m. and 5 p.m. The bacterial suspension of *A. hydrophila* was adjusted to 5.0 × 10^7^ CFU/mL with 0.85% sterile saline. Intraperitoneal injection was used in this artificial challenge [29,31]. After 1-week pre-feeding, each fish in the positive control group and dietary carvacrol group was intraperitoneally injected with 200 µL of bacterial suspension, while the negative control group was intraperitoneally injected with an equal volume of 0.85% sterile saline. Challenged fish were observed daily, and the mortality was recorded for 5 days.

### 2.11. Statistical Analysis

Statistical analyses of the differences between each group were performed with one-way ANOVA using Tukey’s multiple comparison posttest using SPSS 20.0 software. Data are presented as the mean ± standard error (SE) of three independent experiments. The survival rate was analyzed using the Kaplan–Meier estimate method, and the significance of different groups was analyzed with the log-rank test. A *p*-value < 0.05 was considered to indicate a statistically significant difference.

## 3. Results

### 3.1. Inhibitory Effect of Carvacrol on A. hydrophila NJ-35

The MIC and MBC of carvacrol against *A. hydrophila* NJ-35 were 125 µg/mL and 250 µg/mL, respectively. As shown in Figure 1A, carvacrol at sub-MICs (1/4 MIC, 1/8 MIC, 1/16 MIC, 1/32 MIC, and 1/64 MIC) had no significant influence on the growth of *A. hydrophila* NJ-35 (*p* < 0.05). However, the 1/2 MIC of carvacrol exhibited weak inhibitory activity against *A. hydrophila* NJ-35. Furthermore, the sub-inhibitory concentrations were selected to study the effect of carvacrol on the virulence of *A. hydrophila*.

### 3.2. Synergistic Effect of Carvacrol Combined with Antibiotics

Regarding the synergistic potential of carvacrol in combination with antibiotics, the results showed that carvacrol had no antagonistic effect with any of the tested antibiotics. Carvacrol combined with neomycin sulfate showed an additive effect on *A. hydrophila* NJ-35 (FICI = 0.563, Appendix A). The indifferent effect was presented along with enrofloxacin, florfenicol, doxycycline hyclate, and thiamphenicol (FICI = 1.100, 1.063, 1.062, and 1.500, respectively). Generally, carvacrol had no significant synergistic effect with the antibiotics tested in this study; however, a 2- to 16-fold decrease in the MIC of the antibiotics was documented in the synergy tests (Appendix A).

### 3.3. Antibiofilm Activity of Carvacrol

The inhibition of carvacrol at sub-MICs (1/4 MIC, 1/8 MIC, 1/16 MIC, 1/32 MIC, and 1/64 MIC) on the biofilm formation of *A. hydrophila* NJ-35 was measured using crystal violet biofilm assays (Figure 1B). The presence of DMSO did not significantly affect the biofilm formation of *A. hydrophila* NJ-35 (*p* > 0.05). In contrast, carvacrol significantly inhibited *A. hydrophila* biofilm formation at all tested concentrations (*p* < 0.05); this inhibitory effect was significantly enhanced when co-cultured with carvacrol at concentrations of 1/8 MIC and 1/4 MIC (*p* < 0.05).

Carvacrol demonstrated a significant biofilm eradication effect on *A. hydrophila* NJ-35 (Figure 1C). At 6 h and 24 h of treatment, carvacrol showed a killing effect on mature biofilms at a sub-inhibitory concentration (1/4 MIC). Moreover, the biofilm eradication effect increased with treatment time and carvacrol concentration.

Light microscopic observation further confirmed the antibiofilm potential of carvacrol against *A. hydrophila*. In the control group, a well-structured biofilm matrix was observed on the slides, while in treated groups (1/16 MIC, 1/4MIC, and MIC), the biofilm-forming cells and biofilm-covered surface area showed dose-dependent attenuation (Figure 2).

### 3.4. Quantification of EPS Production

At sub-inhibitory concentrations, EPS quantification revealed that carvacrol significantly reduced EPS production by *A. hydrophila*, and the maximum reduction was observed at a 1/4 MIC dose (Figure 1D) compared to the untreated control (*p* < 0.05). Additionally, DMSO (negative control) did not significantly affect the EPS production of *A. hydrophila* NJ-35 compared to the control (*p* > 0.05).

### 3.5. Effect of Carvacrol on Protease and Hemolytic Activities

Carvacrol markedly decreased the protease production of *A. hydrophila* NJ-35 in a dose-dependent pattern (Figure 3). The protease activity was the lowest in the 1/4 MIC treatment group. Carvacrol at ≤1/8 MIC did not impact the hemolytic activity of *A. hydrophila* NJ-35 (*p* > 0.05); however, 1/4 MIC carvacrol significantly inhibited the hemolytic activity (*p* < 0.05).

### 3.6. Modulation of A. hydrophila Virulence Gene Expression by Carvacrol

Carvacrol dramatically reduced the expression of *A. hydrophila* NJ-35 virulence genes (Figure 4). Compared to the control, carvacrol (1/4 MIC, 1/64 MIC) treatment obviously downregulated expression of *flaB* (0.60-fold, 0.49-fold), *aha* (0.47-fold, 0.71-fold), *ompA* (0.30-fold, 0.59-fold), *act* (0.52-fold, 0.49-fold), *aerA* (0.63-fold, 0.52-fold), and *hly* (0.53-fold, 0.50-fold). Additionally, the 1/4 MIC carvacrol downregulated expression of *ahp* (0.57-fold), *ela* (0.07-fold), and *AhyR* (0.86-fold).

### 3.7. Protective Effects of Carvacrol on Grass Carp against A. hydrophila Infection

The artificial challenge test showed that carvacrol increased the survival of grass carp with *A. hydrophila* infection, while no death was observed in the negative group (Figure 5). At 5 days post injection, all of the fish in the basal diet and carvacrol groups were dead. The survival rate in the positive group was 24.00%, while that in the carvacrol group was 56.00%. The survival rate in the carvacrol group was significantly increased compared to that in the positive control group (log-rank test, *p* < 0.05). The moribund grass carp showed hemorrhagic septicemia symptoms, and bacteria isolated from diseased fish (spleen and kidney) were confirmed as *A. hydrophila.*

## 4. Discussion

Bacterial septicemia caused by *A. hydrophila* infection is a serious threat to healthy aquaculture [19]. However, the use of antibiotics against this disease has attracted increasing awareness of drug resistance and food safety. Phytochemicals possess good antibacterial activity, suggesting that they may be an alternative to antibiotics against bacteria such as *A. hydrophila*, *A. sobria*, *Citrobacter freundii,* and *Raoultella ornithinolytica* [24,33].

Carvacrol is an antimicrobial drug with potential microbiological activities against fish bacterial pathogens [34,35]. Bandeira Junior et al. (2018) reported that the MIC and MBC of carvacrol against *A. hydrophila* ATCC 7966 were 100 µg/mL and 200 µg/mL, respectively, which was similar to our results [33]. However, the mechanism underlying the antibacterial action of carvacrol has not been fully elucidated. One potential explanation is that carvacrol can disrupt bacterial envelopes. Previous studies have revealed that carvacrol could damage the integrity of the bacterial cell membrane, causing bacterial lysis, leakage of cytoplasmic contents, and even death [36,37]. Notably, the high antibacterial activity of carvacrol is attributed to the presence of a polar functional group [38]. In addition, studies have shown that the intracellular targets of essential oil may be related to its antimicrobial properties [39]. However, the target of carvacrol against *A. hydrophila* remains unclear and requires further investigation.

Recent research has shown that the inhibition of virulence is a promising strategy against pathogenic bacterial infection [40]. Such an anti-virulence therapy could consist of either inhibiting certain virulence factors (e.g., biofilm, hemolysin, and protease) or, specifically, interfering with the regulation of virulence factor expression (e.g., quorum sensing system).

Biofilm formation is a process by which microbial cells aggregate to form collectives embedded in a self-produced extracellular matrix [41]. Biofilms improve the capability of bacteria to combat antimicrobials, hence increasing their harm to the host immune system and diminishing the effectiveness of antimicrobials [42,43]. Therefore, inhibiting biofilm formation is a crucial method for impeding bacterial infections. Studies have demonstrated that carvacrol can inhibit bacterial biofilm formation at sub-MICs, with a stronger impact reported at higher concentrations [44,45]. Liu et al. (2021) reported that the biofilm formation of *Enterobacter cloacae* was inhibited by carvacrol at sub-inhibitory concentrations of 64 and 128 µg/mL, and carvacrol decreased biofilm thickness and extrapolymeric matrix excretion, as evidenced by microscopic investigations [7]. In this research, carvacrol significantly inhibited the formation of *A. hydrophila* NJ-35 biofilms at sub-MICs, with an increased inhibitory effect observed with increasing carvacrol concentration. This result was also confirmed by microscopic observation. Moreover, the treatment of mature biofilms is more challenging than that of early stage biofilms and exhibits increased drug resistance. After culturing *A. hydrophila* for 24 h, the biofilms appeared to reach a maximum density, after which no further increase occurred [46]. In this study, we demonstrated a strong biofilm eradication effect of carvacrol on *A. hydrophila* NJ-35.

EPS is the extrapolymeric matrix component of bacteria encapsulated in the biofilm [47]. During colonization, *A. hydrophila* could produce extracellular polymeric substances, thereby resulting in the development of mature biofilms. Hence, biofilms lacking EPS barriers are more likely to expose bacteria to the drug action and host immune system. In this study, the EPS production was apparently inhibited by carvacrol at sub-MICs. Similarly, carvacrol inhibited *A. hydrophila* NJ-35 biofilms at sub-MICs in a manner that correlated well with the attenuated production of EPS.

Hemolysins and proteases are the crucial extracellular virulence factors produced by *A. hydrophila* [48]. By damaging host tissues, these enzymes allow pathogens to gain nutrients and spread [49]. We observed that carvacrol has a concentration-dependent effect on the synthesis of proteases by *A. hydrophila* NJ-35. However, carvacrol could only inhibit the hemolysis of *A. hydrophila* NJ-35 until the carvacrol concertation reached 1/4 MIC. Conversely, carvacrol could decrease the hemolytic activity of *A. hydrophila* MF 372510 at sub-inhibitory concentrations [33]. This discrepancy may be due to the different strains of *A. hydrophila* used.

The adhesion and toxin of *A. hydrophila* encoded by *aha*, *flaB*, and *ompA* genes are located in the outer cell membrane and play an important role in maintaining cytoskeletal structure, biofilm formation, nutrition transport, and resistance to host immune systems [50,51]. In this study, the sub-inhibitory doses of carvacrol exhibited a substantial downregulatory impact on *aha, flaB,* and *ompA*. This result was consistent with anti-biofilm efficacy of carvacrol against *A. hydrophila*. The *aha* and *ela* genes are mainly responsible for regulating extracellular proteases secreted by *A. hydrophila*. Serine protease has caseinolytic activity, and elastase has both elastolytic and caseinolytic activity [52,53]. Here, we found that carvacrol downregulated both the transcription and translation of *ahp* and *ela* genes in *A. hydrophila* NJ-35. This suggests that carvacrol triggered the downregulation of pathogenicity-related genes, thereby reducing the virulence and pathogenicity of *A. hydrophila*. In addition, toxin genes encoded by the *act* and *hly* have been used for assessing potential pathogenesis of *A. hydrophila*. Aerolysin is a functional enzyme, which is extensively homologous to enterotoxins. Studies have demonstrated that thymol could protect the channel catfish from *A. hydrophila* infection by inhibiting the transcription of the *aerA* gene [40]. Our results also indicated that carvacrol reduced *act*, *aerA*, and *hly* genes expression.

Quorum sensing (QS) systems are signaling networks that regulate bacterial behavior, virulence, and biofilm formation [54]. The biofilm development, extracellular protease, and hemolysin production of *A. hydrophila* are positively controlled by the QS regulatory protein AhyR [55]. Here, carvacrol was found to downregulate *ahyR* gene expression of *A. hydrophila*, suggesting its involvement in QS-mediated virulence factor expression.

In this study, the carvacrol-supplemented diet increased the resistance of grass carp to *A. hydrophila* infection. Similar results have also been observed in *Colossoma macropomum*, *Ictalurus punctatus*, and *Carassius auratus* [56,57,58]. Besides the antibacterial capability, Silva et al. (2021) found that tambaqui (*C. macropomum*) fed with carvacrol had higher monocyte and neutrophil counts, phagocytic activity, and a higher survival rate when exposed to *A. hydrophila* infection [56]. Thus, we inferred that carvacrol could enhance the non-specific cellular immune function of grass carp and improve their ability to resist bacterial invasion.

Essential oils are commonly considered to have multi-target inhibitory effects on pathogenic bacteria and, in combination with conventional antibiotics, may enhance the activity, avoid the emergence of antibiotic resistance, and reduce drug use [59,60]. Pirog et al. (2019) found that essential oils combined with other antimicrobials could destroy yeast and bacterial biofilm, thereby significantly decreasing their MIC [61]. Bandeira Junior et al. (2019) also found that carvacrol had an additive effect with florfenicol on *A. hydrophila* [62]. In this investigation, the antibacterial activity of carvacrol combined with neomycin sulfate showed an additive effect, suggesting that a combination of conventional antibiotics with carvacrol is a promising alternative for the control of *A. hydrophila* infection in aquaculture.

## 5. Conclusions

In conclusion, our results demonstrate that carvacrol has antimicrobial and anti-virulence activities against *A. hydrophila* NJ-35. The sub-inhibition concentration of carvacrol could inhibit protease production, hemolytic activity, EPS production, and biofilm formation. Meanwhile, carvacrol inhibited the transcription of virulence genes. However, carvacrol showed no antagonistic effect with antibiotics commonly used in aquaculture, and supplementation with carvacrol in diet could also increase the survival of grass carp infected with *A. hydrophila*.

## Figures and Tables

**Figure 1 microorganisms-10-02170-f001:**
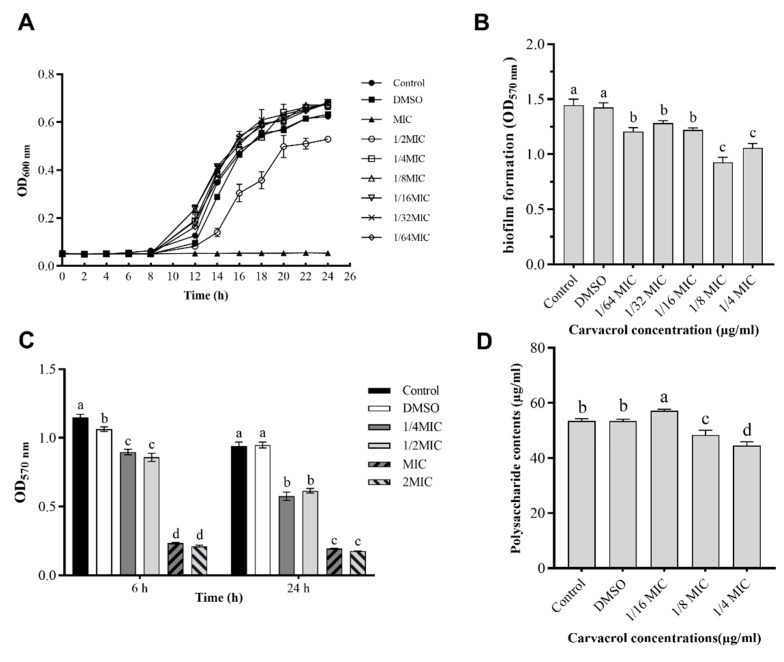
Antibiofilm activity of carvacrol against *Aeromonas hydrophila*. (**A**). The effect of carvacrol on *A. hydrophila* growth according to MIC and sub-MICs. Carvacrol concentration: Control (0), MIC (125 µg/ml), 1/2 MIC (62.5 µg/ml), 1/4 MIC (31.25 µg/ml), 1/8 MIC (15.625 µg/ml), 1/16 MIC (7.8125 µg/ml), 1/32 MIC (3.90625 µg/ml), and 1/64 MIC (1.953125 µg/ml). (**B**). The effect of carvacrol on biofilm formation in *A. hydrophila*. (**C**). The effect of carvacrol on preformed biofilms of *A. hydrophila*. (**D**). The effect of carvacrol on exopolysaccharide (EPS) production by *A. hydrophila*. Data are presented as the mean ± standard error (SE) of three independent experiments. The results were analyzed with one-way ANOVA using Tukey’s multiple comparison posttest. a–d: Values with different letters are significantly different (*p* < 0.05), while those with similar letters are not. MIC, minimum inhibitory concentration.

**Figure 2 microorganisms-10-02170-f002:**
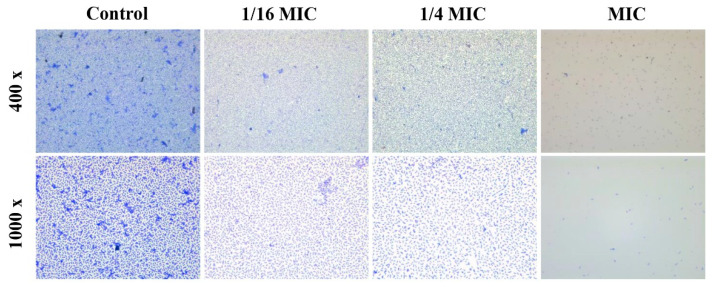
Microscopic analyses of *A. hydrophila* biofilm formation treated with carvacrol. The images were taken at magnifications of 400× and 1000×.

**Figure 3 microorganisms-10-02170-f003:**
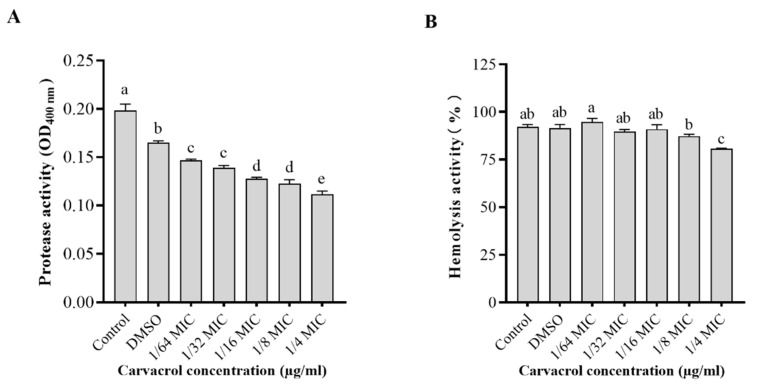
Effect of carvacrol on protease (**A**) and hemolysis (**B**) activities in *A. hydrophila*. Data are presented as the mean ± standard error (SE) of three independent experiments. The results were analyzed with one-way ANOVA using Tukey’s multiple comparison posttest. a–e: Values with different letters are significantly different (*p* < 0.05), while those with similar letters are not.

**Figure 4 microorganisms-10-02170-f004:**
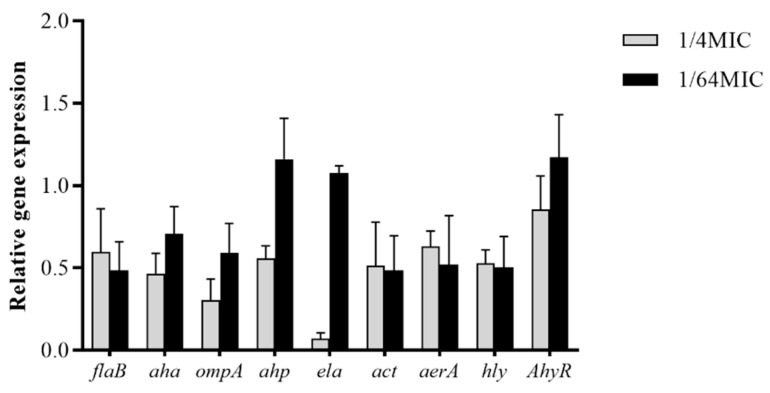
Effect of carvacrol on the virulence gene expression in *A. hydrophila.* Data are shown as the value of the carvacrol treated (1/4 MIC, 1/64 MIC) group divided by that of the DMSO (negative control) group. Data are presented as the mean ± standard error (SE) of three independent experiments.

**Figure 5 microorganisms-10-02170-f005:**
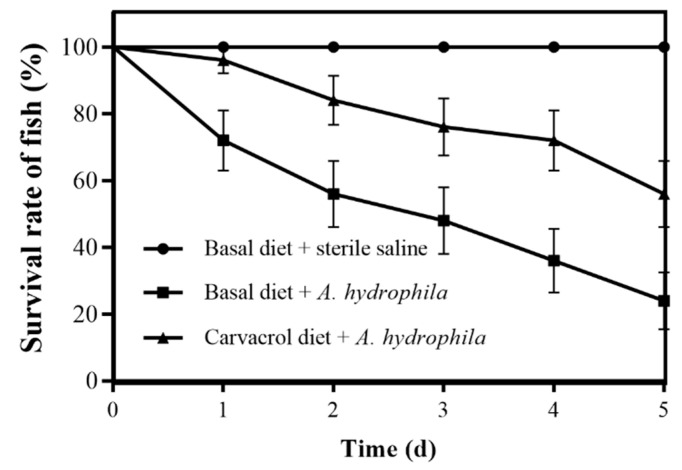
Effect of dietary carvacrol on the survival of grass carp infected with *A. hydrophila* NJ-35. In the basal diet and carvacrol groups, 25 fish were intraperitoneally injected with *A. hydrophila* or sterile saline, respectively. Error bars: 95% confidence intervals.

## Data Availability

Data are available upon request.

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
