# Peer review of "Antimicrobial and Antivirulence Activities of Carvacrol against Pathogenic Aeromonas hydrophila"

_microorganisms, 2022, doi:10.3390/microorganisms10112170_

Round 1

Reviewer 1 Report

Dear Authors!

My comments are in attached file.

Author Response

Response to Reviewer 1 Comments

General comment
The manuscript by Wang et al describes a systematic and detailed study of various aspects of the antibacterial activity of carvacrol against the pathogen of Aeromonas hydrophila. The studies were performrd at the proper methodological level, the results are substantiated and well discussed with a logical conclusion.

Point 1. Table 1 can be transferred to the supplementary department, this information does not apply to the actual results of the article, but has a link to another study.

Response: Thank you for your comment. We have included Table 1 in the supplementary materials (now Table S1).

Point 2: All the figures are of low resolution, I would recommend to replace them.

Response: Thank you for your kind reminder. We have revised it in the manuscript and resubmitted the figures with high resolution.

Point 3: Figure 6 - no confidence intervals.

Response: Thank you for your kind reminder. We have revised it in the manuscript and added confidence intervals (95% CI) in Figure 6.

Point 4: Line 125 ddH2O - bidistilled water? It requires decryption. Line 150 – EPS.

Response: Thank you for your kind reminder. We have revised it in the manuscript (double-distilled water, ddH2O; EPS, exopolysaccharide). Please refer to lines 117, 143.

We would like to thank the referee again for taking the time to review our manuscript.

Reviewer 2 Report

In this paper Wang et al. reports on antibacterial properties of carvacrol on Aeromonas hydrophilia and anti infective activity in an in vivo carp model. The results support and add to many previous studies on the antimicrobial properties of carvacrol, in particular through the in vivo results.

Overall the approach taken is scientifically sound, but the paper has significant shortcomings both concerning data presentation and analysis. In addition, the paper is poorly written and requires thorough revision by a native English speaking scientist.

Specifically:

1.       The title “promises”  “antimicrobial efficacy combined with antibiotics”. However, since no synergy is found there is no positive effect with antibiotics, and thus this should not appear in the title.

2.       Line 31-32: Aeromonas hydrophila is bacterium not a disease!

3.       Line 47: The cytotoxic effects of the antibiotics as such are not reduced, but the dose can be reduced, thus reducing the side effects of the treatment.

4.       Line 59-60: “it is essential to examine its effect”. There is no argument for “essential”.

5.       Line 197: It should be explained why an ip infection model was chosen and what the advantages are of this.

6.       Lines 212-221. The MIC assay is the standard initial method of measuring antibacterial activity, thus there is no reason to present a detailed description. Basically a MIC value would suffice.

7.       Line 220: carvacrol does not have a POWERFUL inhibitory impact. It is two orders of magnitude less active than most standard antibiotics.

8.       Figure 1 data: Significance parameters must be provided, including number of biological replicates for the data.

9.       Table 2/Figure2: Since no synergy is found these data should be moved into supporting data, and the two graphs of Figure 2, present identical data, thus one of them should be deleted.

10.   Line 258: “decreased by 15.37%, 35.82%, and 26.83%”: The data does not support this accuracy at all.

11.   Figure 4: Significance parameters must be provided, including number of biological replicates for the data.

12.   Figure 4B: Why was data for ½ MIC not included? This can strengthen the conclusion.

13.   Figure 5: Number of biological replicates for the data must be included

14.   Figure 5: A dose response would strengthen the conclusions

15.   Lines 329-330: “fish infections such as A. hydrophila, A. sobria, Citrobacter freundii, and Raoultella ornithinolytica”: These are bacteria not infections.

16.   Lines 334- 339. The growth curves (as shown) reflect standard behavior in MIC assays, and do not provide special information.

17.   Line 363: What is “on light micrographs”?

18.   Line 439: “carvacrol has high antimicrobial and anti-virulence activities”. The activity is not high, it is 100 fold lower than most other antibiotics in use.

19.   Line 344-5: carvacrol has only one hydroxyl group.

20.   Line 447-8: “It is imperative to perform future studies involving making it more applicable through nanotechnology and dose-response to meet clinical needs”. This sentence has basically no real content, and clinical needs are predominantly connected to human patients, and fish are not accepted models for gaining information on human diseases.    

Author Response

Response to Reviewer 2 Comments

General comment
In this paper, Wang et al. reports on antibacterial properties of carvacrol on Aeromonas hydrophilia and anti-infective activity in an in vivo carp model. The results support and add to many previous studies on the antimicrobial properties of carvacrol, in particular through the in vivo results.

Point 1. Overall the approach taken is scientifically sound, but the paper has significant shortcomings both concerning data presentation and analysis. In addition, the paper is poorly written and requires thorough revision by a native English speaking scientist.

Response: Thank you for your suggestion. We have revised the data presentation and improved the language expression in the manuscript.

Point 2. The title “promises”  “antimicrobial efficacy combined with antibiotics”. However, since no synergy is found there is no positive effect with antibiotics, and thus this should not appear in the title.

Response: Thank you for your comment. Since the original title could cause confusion, we have modified the title of the manuscript to “Antimicrobial and antivirulence activities of carvacrol against pathogenic Aeromonas hydrophila”. We hope our modification could meet with your concerns.

Point 3: Line 31-32: Aeromonas hydrophila is bacterium not a disease!

Response: Thank you for your kind reminder. We have revised it in the manuscript. Please refer to line 31. (“Aeromonas hydrophila, a common gram-negative pathogenic bacterium, is ubiquitously dispersed in freshwater environment and has been considered as an important opportunistic pathogen of fish, amphibians, reptiles, and mammals.”)

Point 4: Line 47: The cytotoxic effects of the antibiotics as such are not reduced, but the dose can be reduced, thus reducing the side effects of the treatment.

Response: Thank you for your kind reminder. We totally agree with your opinion. We have revised it in the manuscript. Please refer to line 44. (“The previous studies have demonstrated that the use of anti-virulence medications in conjunction with antibiotics can boost the effectiveness of the antibiotics and reduce the dosage needed.”)

Point 5: Line 59-60: “it is essential to examine its effect”. There is no argument for “essential”.

Response: Thank you for your suggestion. We have revised it in the manuscript. Please refer to line 55.

Point 6: Line 197: It should be explained why an ip infection model was chosen and what the advantages are of this.

Response: Thank you for your advice. Intraperitoneal injection is one of the more common infection models. Compared with immersion and oral gavage, it has the characteristics of high infection intensity, fast speed, and easy operation. We have added relevant references in the manuscript. Please refer to line 186.

Point 7: Lines 212-221. The MIC assay is the standard initial method of measuring antibacterial activity, thus there is no reason to present a detailed description. Basically a MIC value would suffice.

Response: Thank you for your suggestion. We have revised it in the manuscript. Please refer to lines 201-207.

Point 8:  Line 220: carvacrol does not have a POWERFUL inhibitory impact. It is two orders of magnitude less active than most standard antibiotics.

Response: Thank you for your suggestion. We have revised it in the manuscript.

Point 9: Figure 1 data: Significance parameters must be provided, including number of biological replicates for the data.

Response: Thank you for your advice. We added the number of biological replicates in Figure 1, Figure 4, and Figure 5 (now Figure 1, Figure 3, and Figure 4).

Point 10: Table 2/Figure2: Since no synergy is found these data should be moved into supporting data, and the two graphs of Figure 2, present identical data, thus one of them should be deleted.

Response: Thank you for your suggestion. We totally agree with your opinion. We have revised it in the manuscript (now Table S2 and Figure S1 ).

Point 11: Line 258: “decreased by 15.37%, 35.82%, and 26.83%”: The data does not support this accuracy at all.

Response: Thank you for your kind reminder. To calculate the biofilm inhibition (%) in all treated wells, the following formula was used:

Biofilm formation inhibition % (%) was defined as [(OD of untreated control − OD of treated sample) × 100]/OD of untreated control.

The formula also applies to biofilm eradication and protease activity. We have added corresponding explanations in the manuscript. Please refer to line 122.

Point 12: Figure 4: Significance parameters must be provided, including number of biological replicates for the data.

Response: Thank you for your advice. We added the number of biological replicates in Figure 1, Figure 4, and Figure 5 (now Figure 1, Figure 3, and Figure 4).

Point 13: Figure 4B: Why was data for ½ MIC not included? This can strengthen the conclusion.

Response: Thank you for your suggestion. The antimicrobial concentrations that did not inhibit bacterial growth were chosen as the subinhibitory concentrations and used for the subsequent phenotypic virulence assays. The growth curve analysis showed that 1/2 MIC of carvacrol inhibited the growth of A. hydrophila. Therefore, the 1/2 MIC of carvacrol was not chosen to use as the tested concentration.

Point 14: Figure 5: Number of biological replicates for the data must be included

Response: Thank you for your advice. We added the number of biological replicates in Figure 1, Figure 4, and Figure 5 (now Figure 1, Figure 3, and Figure 4).

Point 15: Figure 5: A dose response would strengthen the conclusions

Response: Thank you for your suggestion. We totally agree with your opinion. We added a low-dose treatment group (1/64 MIC) in Figure 5 (now Figure 4). Please refer to line 265.

Point 16: Lines 329-330: “fish infections such as A. hydrophila, A. sobria, Citrobacter freundii, and Raoultella ornithinolytica”: These are bacteria not infections.

Response: Thank you for your kind reminder. We have revised it in the manuscript. Please refer to line 293. (“Phytochemicals have been reported to possess good antibacterial activity, indicating that they are a possible alternative to antibiotics against bacteria such as A. hydrophila, A. sobria, Citrobacter freundii, and Raoultella ornithinolytica.”)

Point 17: Lines 334- 339. The growth curves (as shown) reflect standard behavior in MIC assays, and do not provide special information.

Response: Thank you for your suggestion. We have revised it in the manuscript.

Point 18: Line 363: What is “on light micrographs”?

Response: Thank you for your kind reminder. We have revised it in the manuscript. Please refer to line 325. (“This result was also confirmed with microscopic observation.”)

Point 19: Line 439: “carvacrol has high antimicrobial and anti-virulence activities”. The activity is not high, it is 100 fold lower than most other antibiotics in use.

Response: Thank you for your kind reminder. We have revised it in the manuscript. Please refer to line 386.

Point 20: Line 344-5: carvacrol has only one hydroxyl group.

Response: Thank you for your advice. Hydroxyl groups, commonly found in thymol, eugenol, terpineol and carvacrol, are highly reactive and can form hydrogen bonds with active sites of target enzymes, which inactivates proteins and leads to dysfunction or rupture of the cell membrane[1]. Studies have shown that compounds with low molecular weight and polar functional groups (such as Ar-OH) can increase antibacterial activity by facilitating penetration through the outer cell membrane[2]. We have revised the text to address your concerns and hope that it is now clearer. Please refer to line 304.

[1] Kim, J.; Marshall, M.R.; Wei, C. Antibacterial activity of some essential oil components against five foodborne pathogens. J. Agric. Food Chem. 1995, 43, 2839–2845.

[2] Guimarães, A.C.; Meireles, L.M.; Lemos, M.F.; Guimarães, M.C.C.; Endringer, D.C.; Fronza, M.; Scherer, R. Antibacterial activity of terpenes and terpenoids present in essential oils. Molecules 2019, 24, 2471.

Point 21: Line 447-8: “It is imperative to perform future studies involving making it more applicable through nanotechnology and dose-response to meet clinical needs”. This sentence has basically no real content, and clinical needs are predominantly connected to human patients, and fish are not accepted models for gaining information on human diseases.

Response: Thank you for your suggestion. We have revised it in the manuscript.

We would like to thank the referee again for taking the time to review our manuscript.

Round 2

Reviewer 2 Report

Although the ms has been improved, some issues still remain.

1.       It should be fundamental scientific knowledge that data reporting in terms of number of digits must reflect the accuracy of the data. This is not the case in the legend to Fig. 1, in lines 235-6, 247 and 252

2.       The statistics presented in Figures 1B, 1D and Figure 3 are still unclear. Which samples are statistically different and what are the p values for these differences? The letter code (a, b, c, d)  is not defined

3.       Line 294: Carvacrol is not a (certified) drug, it is a natural product, and it is an overstatement to describe it as potent.

4.       Line 299: I do not believe that the authors mean “PREVIOUS researchers”.

5.       Line 302: Carvacrol only has one polar functional group.

6.       Line 335: There are many “haemolysins and proteases”.

7.       In general, the writing still needs improvement (by a native English speaking scientist in the field).

Author Response

Response to Reviewer 2 Comments

Point 1. It should be fundamental scientific knowledge that data reporting in terms of number of digits must reflect the accuracy of the data. This is not the case in the legend to Fig. 1, in lines 235-6, 247 and 252.

Response: Thank you for your suggestion. We have revised the data presented in the “Results” section. Please refer to lines 228, 247, and 257.

Point 2. The statistics presented in Figures 1B, 1D and Figure 3 are still unclear. Which samples are statistically different and what are the p values for these differences? The letter code (a, b, c, d)  is not defined.

Response: Thank you for your kind reminder. Value with different letters (a, b, c, d) are significantly different (P < 0.05), while those with similar letters are not significantly different. We have added the corresponding instructions in Figure 1 and Figure 3.

Point 3: Line 294: Carvacrol is not a (certified) drug, it is a natural product, and it is an overstatement to describe it as potent.

Response: Thank you for your kind reminder. We have revised it in the manuscript. Please refer to line 294. (“Carvacrol is an antimicrobial drug with potential microbiological activities against fish bacterial pathogens.”)

Point 4: Line 299: I do not believe that the authors mean “PREVIOUS researchers”.

Response: Thank you for your kind reminder. We have revised it in the manuscript. Please refer to line 299. (“Previous studies have revealed that…”)

Point 5:  Line 302: Carvacrol only has one polar functional group.

Response: Thank you for your kind reminder. We have revised it in the manuscript. Please refer to line 302. (“…a polar functional group.”)

Point 6:  Line 335: There are many “haemolysins and proteases”.

Response: Thank you for your kind reminder. We have revised it in the manuscript. Please refer to line 335.

Point 7: In general, the writing still needs improvement (by a native English speaking scientist in the field).

Response: Thank you for your suggestion. We have asked a native speaker to modify and improve the language for the manuscript.

We would like to thank the referee again for taking the time to review our manuscript.
